# CircEZH2 Regulates Milk Fat Metabolism through miR-378b Sponge Activity

**DOI:** 10.3390/ani12060718

**Published:** 2022-03-12

**Authors:** Dongyang Wang, Zhengjiang Zhao, Yiru Shi, Junyi Luo, Ting Chen, Qianyun Xi, Yongliang Zhang, Jiajie Sun

**Affiliations:** College of Animal Science, Guangdong Provincial Key Laboratory of Animal Nutrition Control, National Engineering Research Center for Breeding Swine Industry, South China Agricultural University, Guangzhou 510642, China; wangdy960821@outlook.com (D.W.); orcsblade@163.com (Z.Z.); shiyiru00@163.com (Y.S.); luojunyi@scau.edu.cn (J.L.); allinchen@scau.edu.cn (T.C.); xqy0228@163.com (Q.X.)

**Keywords:** circEZH2, proliferation, apoptosis, miR378b, fatty acid metabolism

## Abstract

**Simple Summary:**

Heat stress has seriously threatened the performance and health of dairy cows and has become one of the most important factors restricting the development of the dairy industry. In our previous study, we found that heat stress markedly altered the expression patterns of circRNAs in dairy cow’s mammary gland tissue, and heat-induced circRNAs participated in the regulation of milk fat metabolism through competing endogenous RNA (ceRNA) networks. Therefore, we evaluated the roles of heat-induced circEZH2 in the regulation of milk fat metabolism in this study. In more detail, we found that circEZH2 affects the proliferation, apoptosis, and lipid metabolism of mammary gland epithelial cells, and successfully verified the targeting relationship of circEZH2-bta-miR378b-LPL and circEZH2-bta-miR378b-CD36. This experiment expands the basic data on the role of circRNA in milk fat regulation, and provides a theoretical basis for alleviating heat stress in dairy cows.

**Abstract:**

In this study, we evaluated the roles of heat-induced circEZH2 in the regulation of milk fat metabolism. CircEZH2 overexpression increased HC11 cell proliferation and decreased apoptosis. These changes were accompanied by increased expression of proliferation marker proteins (PCNA, Cyclin D, and Cyclin E) and the anti-apoptotic protein Bcl2, while expression of the pro-apoptotic proteins Bax and cleaved-caspase was reduced. SiRNA-mediated silencing of EZH2 in HC11 cells had the opposite effects. CircEZH2 overexpression promoted the uptake of a fluorescent fatty acid (Bodipy) as well as expression of the fatty acid transport-related protein CD36, lipolysis-related protein LPL, and unsaturated fatty acid metabolism-related proteins FADS1 and SCD1. Dual luciferase reporter assays verified the targeting relationship of the two ceRNA networks, circEZH2-miR378b-LPL and circEZH2-miR378b-CD36. This information provides further clarification of the role of circRNAs in milk fat regulation in addition to a theoretical basis for alleviating the effects of heat stress on milk production by dairy cows.

## 1. Introduction

Milk is a natural and healthy drink, favored for its rich nutritional value, unique flavor, and outstanding health functions. As one of the major nutrients in milk [1], milk fat not only affects the flavor and nutritional value of milk, but also participates in the process of nutrient metabolism that regulates human growth and development [2]. Therefore, exploration of mechanisms regulating milk fat synthesis and fatty acid composition is an active area of research [3]. Milk fat metabolism is a complex process regulated by multiple genes as well as a series of regulatory factors [4]. Although some factors have been identified using high-throughput sequencing technology, the underlying regulatory mechanisms remain to be elucidated [5].

Circular RNAs (circRNAs) are naturally occurring, endogenous non-coding RNA (ncRNA) with a closed circular structure [6]. Numerous studies have shown that circRNAs participate in regulating the growth, development, reproduction, and health of livestock, especially in the process of animal lactation [7]. Many circRNAs that are closely related to lactation have been found in the mammary gland tissues of humans [8], dairy cows [9], sheep [10], goats [11], and sows [12]. These studies have shown that circRNAs play important regulatory roles in the development of animal mammary glands, milk secretion, and the synthesis of nutrients in milk. However, the function and mechanism of related circRNAs are still unclear.

In our previous study, we showed that heat stress markedly altered the expression patterns of circRNAs and revealed that as a competitive endogenous circRNA, circEZH2 (circular Enhancer of zeste 2 polycomb repressive complex 2 subunit) participates in the regulation of milk fat metabolism [13]. In this study, we hypothesized that the heat induced circEZH2 plays an important role in the regulation of milk fat metabolism, providing valuable insights into the mechanisms that control lactation and milk quality.

## 2. Materials and Methods

### 2.1. RNA Extraction and Real-Time Quantitative PCR

We extracted the total RNA from the HC11 mouse mammary epithelial cells using TRIzol reagent (Invitrogen, Carlsbad, CA, USA) according to the manufacturer’s instructions, and then reverse-transcribed the RNA into cDNA using the PrimeScript RT Reagent Kit with gDNA Eraser (TaKaRa, Dalian, China). Real-time quantitative PCR (RT-qPCR) of related genes was carried out using the Bio-Rad CFX 96™ Real-Time Detection System and SYBR Green PCR Master Mix (TaKaRa). The formula for calculating relative gene expression was calculated using the 2^−ΔΔCt^ method and the primers are shown in Appendix A. In more detail, qPCR data were derived from six biological replicates, and the gene *glyceraldehyde-3-phosphate dehydrogenase* (*GAPDH*) was used as a reference gene for normalization [9].

### 2.2. Identification of circRNA

To verify the structural homology of mouse and cow circEZH2, we designed a pair of divergent primers that crossed the cyclization site and a pair of convergent primers that did not cross the cyclization site. The authenticity of circEZH2 head and tail splicing was preliminarily verified by electrophoretic separation and Sanger sequencing of the PCR products. To further confirm the presence of circEZH2, we treated total RNA with RNase R enzyme at 37 °C for 10 min and analyzed the RNA expression levels of circular RNA and its linear mRNA by RT-qPCR. The primer sequences used in this experiment are shown in Appendix A.

### 2.3. Plasmid Construction and Oligonucleotide Synthesis

To construct the circEZH2 overexpression plasmid, the full-length sequence of circEZH2 was amplified by adding *BamHI* and *Kpn**I* restriction enzyme sites and protective bases at both ends of the primers. The full-length sequence of circEZH2 was inserted into the pCD2.1-ciR vector (Geneseed, Guangzhou, China) by double enzyme digestion followed by ligation with T4 DNA Ligase (TaKaRa). Based on shared miRNA regulatory elements, two pairs of circEZH2-miRNA-mRNA networks were discovered by miRanda (http://mirtoolsgallery.tech/mirtoolsgallery/node/1055; Accessed date: 5 May 2020): circEZH2-miR-378b-CD36 and circEZH2-miR-378b-LPL. The pmirGLO plasmid (Promega, Madison, WI, USA) was used to design the luciferase reporter constructs containing the miR-378b target sites originating from the 3′-UTR region of *lipoprotein lipase* (*LPL*) and *CD36* genes, and circEZH2. The wild-type (LPL-WT, CD36-WT, and circEZH2-WT), mutant (LPL-MUT, CD36-MUT, and circEZH2-MUT) and deletion (LPL-DEL, CD36-DEL, and circEZH2-DEL) sequences that contained miR-378b binding sites in *LPL*, *CD36*, and circEZH2 were designed and synthesized (Sangon Biotech, Shanghai, China), and restriction enzymes *XbaI* and *XhoI* sequences were added at both ends of the sequence. Finally, these sequences were ligated into the multiple cloning regions in the 3′-UTR of the *Renilla luciferase* gene. The primer sequences used in this experiment are shown in Appendix A. To further explore the function of circEZH2, we also designed and synthesized three interference sequences targeting the circEZH2 junction site, miR-378b mimics (for miRNA overexpression) and miR-378b inhibitor (for miRNA downexpression) (Genepharma, Suzhou, China) as shown in Appendix A.

### 2.4. Cell Culture and Transfection

The HC11 mouse mammary epithelial cell line was purchased from the Chinese Collection of Authenticated Cell Cultures (Beijing, China). After resuscitation, the cells were cultured in a RPMI-1640 medium (Gibco, Grand Island, NY, USA) supplemented with 10% fetal bovine serum (FBS) (Gibco) and 10 ng/mL EGF at 37 °C under 5% CO_2_; the medium was changed every 48 h. Adherent HC11 cells were released from the tissue culture flask by trypsinization and seeded into a 12-well plate (approximately 1 × 10^5^ cells per well). After reaching confluence, the medium was replaced with EGF-free HC11 medium and the cells were cultured for a further 2 days before HC11 differentiation was induced by the addition of 5 µg/mL insulin (Sigma Adlrich, Wisconsin, WI, USA), 100 nM dexamethasone (Sigma) and 5 µg/mL prolactin (Sigma).

At 75% confluence, the stably adherent cells were transfected with plasmid or miR-378b mimics using Lipofectamine 2000 transfection reagent (Invitrogen). After 48 h, the cells were collected for further analysis; three biological replicates were prepared for each treatment.

### 2.5. Western Blot Analysis

HC11 cells (mouse mammary gland cells) were lysed, total proteins were extracted with Radio immunoprecipitation assay (RIPA) reagent (Thermo Scientific, Waltham, MA, USA) containing protease inhibitors, and the protein concentration was determined using the Bicinchoninic acid (BCA) protein assay kit (Nanjing Jiancheng Institute of Bioengineering, Nanjing, China). All extracted proteins were diluted with sodium dodecyl sulfate (SDS) buffer and boiled at 95 °C for 10 min. Equal amounts of proteins were separated by SDS polyacrylamide gel electrophoresis and transferred to a polyvinylidene fluoride (PVDF) membrane (Millipore, Bedford, MA, USA). Membranes were then probed with primary antibodies for the detection of CyclinD, CyclinE, PCNA, Bax, Bcl2, Caspase3, CD36, FADS1, and SCD1. The membranes were then incubated for 30 min at 37 °C with goat anti-rabbit HRP conjugate antibody (Sangon Biotech, Shanghai, China) and goat anti-mouse HRP conjugate antibody (Sangon Biotech). Finally, the protein bands were quantified by densitometry using Image J software V1.8 (National Institutes of Health, Bethesda, MD, USA). All results were expressed as target protein/internal reference protein.

### 2.6. Cell Proliferation Analysis

Cell proliferation was determined by Cell Counting Kit-8 (CCK-8) assay (Bioss, Woburn, MA, USA). Cells were seeded in a 96-well cell culture plate (5 × 10^3^ cells per well), and transfected for 24 h, 48 h, and 72 h. Subsequently, the cells with CCK-8 reagent were then incubated for about 2 h. The optical density (OD) in each well was measured at 450 nm using a multifunctional microplate reader (Bio-Rad, Hercules, CA, USA).

### 2.7. Flow Cytometry Analysis

Two days after transfection, adherent cells were released by digestion with trypsin in the absence of EDTA (Gibco); the digestion reaction was terminated by the addition of RPMI-1640 medium containing 10% FBS. The cell suspension was then centrifuged at 2000× *g* for 5 min at 4 °C. The cell pellet was then washed twice with 1 mL ice-cold 1×PBS (Gibco) and resuspended in RPMI-1640 medium. After counting, Annexin V solution and propidium iodide staining solution were added to the cell suspension (MultiSciences, Hangzhou, China) and incubated for 15 min at 20 °C in the dark. The cell samples were then fixed by incubation overnight with 70% alcohol (Damao, Tianjin, China). After washing the cell pellets with pre-chilled 5% PBS solution, 0.5 mL PI/RNase Staining buffer (Beyotime, Shanghai, China) was added to each sample, and the cells were incubated for 30 min at 37 °C in the dark. Cell cycle analysis was then performed using a CytoFLEX flow cytometer (Beckman Coulter, Miami, FL, USA).

### 2.8. Fluorescent Fatty Acid Uptake Assay

The HC11 cell line was digested and seeded into a 96-well plate (approximately 5 × 10^3^ cells per well). Following stable adherence and culture to 75% confluence, the cells were transfected with the plasmid and cultured for a further 24 h. Before the assay, the cells were starved in serum-free medium for 12 h, and then incubated with Bodipy-C12 (Thermo Fisher Scientific, Bedford, MA, USA) for 5 min at 37 °C without light. After quenching the extracellular fluorescence with trypan blue reagent (Gibco), the absorbance was measured by multifunctional microplate reader and recorded using an inverted fluorescence microscope (FSM-Precision, Suzhou, China), and the value was normalized to CCK8 analysis.

### 2.9. Double Luciferase Activity Analysis

HeLa cells were seeded into a 96-well plate (approximately 5 × 10^3^ cells per well) and cultured in Dulbecco’s Modified Eagle Medium (DMEM) high-glucose medium at 37 °C under 5% CO_2_. Following stable adherence and culture to 75% confluence, the cells were transfected with the reporter gene plasmid vector and miR-378b mimics and cultured for a further 48 h. Luciferase activity was then determined by measuring the absorbance using a multifunctional microplate reader.

### 2.10. Prediction of Translation Function

We used the IRESfinder software (https://github.com/topics/iresfinder; Accessed date: 10 October 2020) to predict the internal ribosome entry site (IRES) region of circEZH2 [14] and searched for IRES sequence activity using the circRNADb database [15]. We then used the ORF_finder software (https://www.ncbi.nlm.nih.gov/orffinder/; Accessed date: 10 October 2020) to predict the coding region of circEZH2, with an ORF sequence of at least 300 bp in length (allowed to span one back-splice joint) [16].

### 2.11. Statistical Analysis

All statistical analysis was performed using SPSS 20.0 software (SPSS Inc., Chicago, IL, USA) and graphs were generated using GraphPad Prism 5.0 software (GraphPad Software Inc., San Diego, CA, USA). Data were expressed as mean and standard deviation (mean ± SD). The differences between treatment groups were analyzed using independent *t*-tests and *p* < 0.05 were set as the thresholds for statistical significance.

## 3. Results

### 3.1. Homology of Mouse CircEZH2 with Bovine Sequences

The mouse circEZH2 sequence was shown to consist of 899 bases encoded by 2–8 exons (Figure 1A). We used cDNA and gDNA isolated from mouse mammary epithelial HC11 cells as templates for amplification of circEZH2 with convergent primers, while successful amplification using cDNA as a template was achieved only using the divergent primers (Figure 1B,C). The circEZH2 circularization site was confirmed by Sanger sequencing (Figure 1D) and found to be similar to the homologous bovine sequence. Generally, circRNA is more resistant to external stimuli due to its ring structure. Linear RNA molecules are susceptible to exonuclease R digestion, while circRNA is resistant. Therefore, RNase R was used to identify the circRNA candidates [17]. Agarose gel electrophoresis and RT-qPCR analysis revealed that there was no significant difference in circEZH2 expression different between the RNase R-treated and mock groups, while the expression of linear parental genes decreased significantly in the treated group (Figure 1E,F), further confirming the successful cyclization of circEZH2.

### 3.2. Construction of the CircEZH2 Overexpression Vector and Interference Sequences

To explore the function of circEZH2, we used specific primers to synthesize the full-length sequence of circEZH2 (Figure 2A) and ligated it into the pCD2.1-ciR vector, which also contains the green fluorescent protein (GFP) expression sequence. Successful construction of the circEZH2 overexpression plasmid (OE-circEZH2) was verified by restriction enzyme digestion (Figure 2B) and Sanger sequencing (Appendix A). Effective transfection of HC11 cells with the empty vector and OE-circEZH2 plasmid was confirmed by detection of GFP protein expression by fluorescence microscopy (Figure 2C). Subsequently, RT-qPCR analysis showed that circEZH2 expression increased significantly in transfected HC11 cells (Figure 2D). In contrast, RT-qPCR analysis showed significant reduction in the expression of circEZH2 in HC11 cells following transfection with the three short interfering RNA (siRNA) sequences compared with the levels detected in cells transfected with the negative control (NC) (Figure 2E). Of these, circEZH2-siRNA1 (si-circEZH2) exhibited the highest interference efficiency and was used in subsequent experiments.

### 3.3. CircEZH2 Affects HC11 Cell Proliferation and Apoptosis

CCK8 assays were used to assess HC11 cell proliferation at 24 h, 48 h, and 72 h after transfection with the OE-circEZH2 plasmid or empty vector. The OD value of the OE-circEZH2 group was significantly higher than that of the empty plasmid group at each time-point, implying that the cell proliferation efficiency was significantly improved (Figure 3A). Flow cytometric analysis also showed that the proportion of cells in the G0/G1 phase was significantly reduced in the OE-circEZH2 group, while the proportion of cells in the S phase and G2/M phase was significantly increased (Figure 3B,C). Overexpression of circEZH2 in HC11 cells promoted the expression of the proliferation marker proteins PCNA, CyclinD, CyclinE, and the anti-apoptotic protein Bcl2, while expression of the pro-apoptotic proteins Bax and Cleaved-caspase 3 was inhibited (Figure 3D,E).

The opposite results were obtained after transfecting HC11 cells with NC and si-circEZH2. CCK8 assays were used to assess HC11 cell proliferation at 24 h, 48 h, and 72 h after transfection with si-circEZH2 or NC. The OD value of the si-circEZH2 group was significantly lower than that in the NC group at each time-point (Figure 4A). Furthermore, flow cytometric cell cycle analysis revealed an increased proportion of cells in G0/G1 phase in the si-circEZH2 group, while the proportion of cells in the S phase and G2/M phase was reduced compared with the NC group (Figure 4B,C). Western blot and qRT-PCR analyses confirmed siRNA-mediated silencing of circEZH2 resulted in significantly increased expression of Bax and Cleaved-caspase 3, while the expression of PCNA, CyclinD, CyclinE, and Bcl2 was inhibited (Figure 4D,E).

### 3.4. CircEZH2 Affects Cell Lipid Metabolism

In our previous study, we showed that heat stress reduced the total amount of oleic acid and unsaturated fatty acids in milk produced by dairy cows, and provided evidence that implicated circEZH2 in milk fat metabolism [13]. Therefore, in the current study, we further explored the role of circEZH2 in milk fat metabolism. Bodipy-C12, which is a red fluorescent derivative of lauric acid, is widely used in the study of fatty acid uptake [18], transport [19], and metabolism [20] in cells. The fluorescence intensity of HC11 cells transfected with OE-circEZH2 was significantly higher than that of the NC group (Figure 5A,B), indicating that circEZH2 overexpression significantly improved the fatty acid uptake capacity of these cells. Western blot and qRT-PCR analyses showed that circEZH2 overexpression significantly promoted the expression of fatty acid metabolism-related proteins (CD36, FADS1, LPL and SCD1) in HC11 cells (Figure 5C,D). The opposite results were obtained following siRNA-mediated silencing of circEZH2. In fluorescent fatty acid uptake assays, the red fluorescence of the si-circRNA group was significantly decreased compared with that in the NC group (Figure 5E,F). Western blot and qRT-PCR analyses confirmed that the expression of fatty acid metabolism-related proteins (CD36, FADS1, LPL and SCD1) was also inhibited (Figure 5G,H).

### 3.5. CircEZH2 Relieves the Negative Effects of Heat Stress

At ambient temperatures, the expression of heat stress proteins in the cell is very low. However, when cells are subjected to heat stress, the concentration of heat stress proteins (especially HSP70) increases rapidly [21]. After exposure of HC11 cells C to 42 °C for 6 h, RT-qPCR analysis showed that *HSP70* expression increased significantly in responded to heat stress (Figure 6A). Using this model, we found that circEZH2 expression also decreased significantly in HC11 cells exposed to heat stress (Figure 6B), which was consistent with our previous observations in vivo [13]. Furthermore, Western blot and qRT-PCR analyses showed that heat stress resulted in decreased expression of PCNA, CyclinD, CyclinE, and Bcl2, while the expression of Bax and Cleaved-caspase increased (Figure 6C,D). Similarly, heat stress also reduced the expression of CD36, FADS1, LPL, and SCD1 proteins in HC11 cells (Figure 6E,F). However, circEZH2 overexpression increased the expression of PCNA, CyclinD, CyclinE, and Bcl2 proteins in HC11 cells under heat stress, while the expression of Bax and Cleaved-caspase 3 was inhibited (Figure 6C,D). In addition, circEZH2 overexpression increased the expression of CD36, FADS1, LPL, and SCD1 proteins in HC11 under heat stress, indicating that it effectively alleviates the adverse effects of heat stress on cell lipid metabolism (Figure 6E,F). These findings indicate that circEZH2 restores the proliferation ability and inhibits apoptosis of HC11 cells under heat stress as well as alleviating the adverse effects of heat stress on lipid metabolism.

### 3.6. Analysis of the CircEZH2 Competitive Regulatory Network

Many studies have demonstrated that circRNAs function as miRNA sponges and play an important role in regulating mRNA expression [22]. Based on shared miRNA regulatory elements, two pairs of circEZH2-miRNA-mRNA networks were discovered: circEZH2-miR-378b-CD36 and circEZH2-miR-378b-LPL (Figure 7A). We then designed and synthesized miR-378b mimics and inhibitors and used RT-qPCR to confirm the efficiency of HC11 cell transfection (Figure 7B). Compared with the NC group, the expression of LPL and CD36 proteins decreased significantly after miR-378b mimics transfection. In contrast, the expression of LPL and CD36 proteins increased significantly after treatment with miR-378b inhibitor (Figure 7C,D). Co-transfection of miR-378b mimics and circEZH2 revealed that circEZH2 effectively alleviated the inhibitory effect of miR-378b mimics on LPL and CD36 expression (Figure 7E,F), thus confirming the existence of a connection between the competitive regulatory networks.

### 3.7. Verification of Target Relationship and Prediction of Translation Ability

We further confirmed the direct target relationships with dual luciferase reporter assays. We designed and synthesized wild-type, mutant, and deleted sequences for circEZH2, LPL, and CD36 that are complementary to the miR-378b sequence and ligated these sequences into the pmirGLO vector. Co-transfection of HeLa cells with the miR-378b mimics and the constructed plasmids showed that miR-378b overexpression significantly reduced the luciferase activity in the circEZH2-WT group, but had no significant effect on the luciferase activity in the circEZH2-DEL and circEZH2-MUT groups (Figure 8A). Using this system, we showed that miR-378b overexpression had similar effects on the luciferase activity of the cells with constructs expressing CD36 (Figure 8B) and LPL (Figure 8C). These results indicated that miR-378b directly regulated circEZH2, LPL and CD36 expression, and confirmed the authenticity of circEZH2-miR-378b-LPL and circEZH2-miR-378b-CD36 ceRNA networks.

These results indicated that circEZH2 directly regulates LPL and CD36 expression via ceRNA networks, although the mechanism underlying the effects on cell proliferation, apoptosis, and unsaturated fatty acid metabolism (FADS1 and SCD1) are still unclear. It has been reported that circRNAs have many potential functions, including alternative splicing of mRNA [23], and interactions with RNA-binding proteins (RBPs) to act as a protein sponge [24]. In recent years, the functions of circRNA-encoded proteins have become a research hotspot [25]. In general, circRNAs lack a 5′-cap and canonical ORF (ORF > 100 amino acids) due to their unique loop structure. Consequently, circRNAs were always thought to be devoid of protein coding ability [26]. However, recent studies have shown that some circRNAs have translation functions and can directly recruit ribosomes to initiate translation through the internal ribosome entry site (IRES) [27]. Here, we identified two IRES fragments on the candidate circEZH2 sequence using the IRESfinder (https://github.com/topics/iresfinder; Accessed date: 10 October 2020). Through ORF finder software (https://www.ncbi.nlm.nih.gov/orffinder/; Accessed date: 10 October 2020) analysis, we found that circEZH2 has an initiation codon and terminator near the splicing site, which can form an ORF (Figure 8D). The target sequence may have coding ability and encode a 290 amino acid circEZH2 polypeptide (Figure 8E).

## 4. Discussion

With the intensification of the global greenhouse effect, heat stress has seriously threatened the performance and health of dairy cows. To date, some studies have indicated that the negative effects of heat stress can be alleviated through physical prevention [28] and nutritional regulation [29]. However, the molecular mechanism of cow lactation has not been fully elucidated. Mammary gland development and activation of lactation are not only regulated by hormones, growth factors, and nutrient supply [30], but also closely related to the genetic changes in the mammals [31]. In recent years, with the rapid development of bioinformatics, it has been discovered that a new type of non-coding RNAs (circRNAs) exists in large amounts in mammary gland tissue [10]. Heat stress has been shown to affect the lactation performance of animals and change the expression of circRNAs in mammary gland tissue [32]. Therefore, it is speculated that circRNAs may be involved in the process of lactation regulation [33].

Numerous circRNAs have been identified in the mammary gland tissue of dairy cows, and changes in expression have been detected during the period of lactation period [9]. Among these, circCSN1S1 expression is positively correlated with milk production and regulates casein secretion through its interaction with miR-2284 [34]. High-throughput sequencing of dairy cow mammary gland tissues in the early and peak periods of lactation revealed that a variety of circRNAs are related to milk fat metabolism and significantly increase the transcription of milk fat synthesis-related genes [35]. For example, circ09863 binds to miR-27a-3p to accelerate triglyceride (TG) accumulation and increase the proportion of unsaturated fatty acids in milk [36]. Although circRNAs have been shown to play a role in lactation, the underlying mechanism is not yet clear. Therefore, in this study, we investigated the changes in the expression of circEZH2 in mammary cells under heat stress further elucidate its role in the mechanism by which milk fats are regulated.

It has been reported that the lactation ability of mammals correlates positively with the number and viability of mammary epithelial cells [37]. Heat stress can destroy the structure of proteins [38], damage the function of cell mitochondria, inhibit proliferation and promote apoptosis of bovine mammary epithelial cells [39]. Therefore, the decrease in milk production and milk quality of dairy cows in summer is probably caused by the inhibitory effects of heat stress on the proliferation of mammary cells and the induction of apoptosis, which, in turn, reduces the number and viability of mammary epithelial cells. In this study, we found that heat stress caused a significant decrease in the expression of cell proliferation-related proteins (PCNA, CyclinD, and CyclinE) and anti-apoptotic protein (Bcl2) in HC11 cells, while the expression of apoptosis-related proteins (Bax and Cleaved-caspase) was significantly increased. These findings are consistent with those of previous studies, and further confirm that high temperature causes serious damage to mammary gland epithelial cells [40]. In this study, we showed that circEZH2 overexpression significantly promoted cell proliferation and inhibited apoptosis, which greatly alleviated the adverse effects of heat stress in HC11 cells.

Generally, the metabolism of fatty acids in mammary gland involves a complex network of reactions, including a series of processes, such as de novo synthesis of fatty acids, uptake and transport of fatty acids, formation of triglycerides, and formation and secretion of lipid droplets, with numerous enzymes playing extremely important roles [41]. In our previous study, we found that heat stress significantly reduced the expression of *LPL*, *CD36*, *SCD1*, and *FADS1* in the mammary glands of dairy cows [13]. As a key enzyme involved in lipid metabolism, LPL hydrolyzes serum triglycerides into glycerol and free fatty acids, which are a key factor in the intake of exogenous fatty acids [42]. CD36 also plays an important role in the process of fatty acid transport. It not only promotes the transmembrane transport of fatty acids, but also regulates the rate of fatty acid uptake by cells [43]. In our previous analysis of milk components, we found that milk fat levels are significantly reduced under heat stress. We speculate that the reduction in the expression of LPL and CD36 caused by heat stress leads to a decrease in fatty acid intake. The fatty acid desaturases SCD1 and the fatty acid deoxygenase FADS1 have a marked influence on the synthesis of unsaturated fatty acids [44,45]. This was confirmed in our previous study of the fatty acid profile of milk. Heat stress causing a decrease in the expression of SCD1 and FADS1, reduced the content of decorating enzymes in the mammary gland, and significantly reduced the content of unsaturated fatty acids in milk [13]. These results were also verified in our cell studies, with heat stress found to significantly reduce the expression of LPL, CD36, SCD1, and FADS1 proteins. Furthermore, through circEZH2 overexpression and siRNA-mediated silencing, we showed that circEZH2 was positively correlated with the expression of CD36, FADS1, LPL, and SCD1. CircEZH2 overexpression also effectively alleviated the adverse effects of heat stress on fat metabolism, which successfully verified the important role of circEZH2 in lipid metabolism.

To date, most studies have shown that circRNAs can function as competitive endogenous RNAs (ceRNAs) by combining with miRNAs to perform their biological functions [46]. Therefore, to explore the mechanism underlying the role of circEZH2 in milk fat metabolism, we analyzed the targeting relationship between circEZH2-miRNA and miRNA-mRNA. We found that the 3′-UTR of *LPL* and *CD36* genes, and the base sequence of circEZH2, contained fragments that are complementary to the seed sequence of miR-378b, and predicted that these might play a novel role in endogenous competitive regulation. It has been reported that miR-378b plays a key role in the regulation of lipid metabolism in rat liver [47]. It has been reported that miR-378 can affect lipogenesis by promoting the expression of fatty acid metabolism-related genes such as *FAS* and *SCD1*, thereby increasing triglyceride production and the size of lipid droplets [48]. In contrast, it has also been speculated that miR-378 promotes lipolysis because its role in lipid metabolism is highly related to catecholamines [49]. This study also showed that miR-378b plays a unique role in lipid metabolism, possibly by regulating the uptake of fatty acids via the ceRNA network. In this study, we successfully verified the target relationship between the two ceRNA networks circEZH2-miR378b-LPL and circEZH2-miR378b-CD36 through using the dual luciferase reporter system. The regulation of these two pairs of ceRNA networks may play an important role in the milk fat metabolism under heat stress.

In general, circRNAs are enriched in miRNA-binding sites that play function as miRNA sponges [50]. However, in this study, we did not predict the miRNA that interacts with ceRNA to explain the effect of circEHZ2 on cell proliferation and apoptosis and unsaturated fatty acid metabolism in HC11 cells. It is speculated that circEZH2 may perform these roles through other functions, such as encoding polypeptides or proteins [51]. Therefore, we used bioinformatics software to predict putative peptides encoded by circEZH2. This analysis indicated that circEZH2 encodes a 298 amino acid polypeptide, which may initiate translation through the IRES site; however, the specific mechanism requires further research.

## 5. Conclusions

We found that HC11 cell proliferation is inhibited, and apoptosis is promoted following exposure to heat stress. Lipoprotein hydrolysis, fatty acid uptake, and unsaturated fatty acid production were also significantly reduced. Thus, we conclude that the heat stress-related circEZH2 effectively alleviates the adverse effects of heat stress in HC11 cells through two ceRNA competitive regulatory networks (circEZH2-miR378b-CD36, circEZH2-miR378b-LPL) to restore cell proliferation, apoptosis, and lipid metabolism.

## Figures and Tables

**Figure 1 animals-12-00718-f001:**
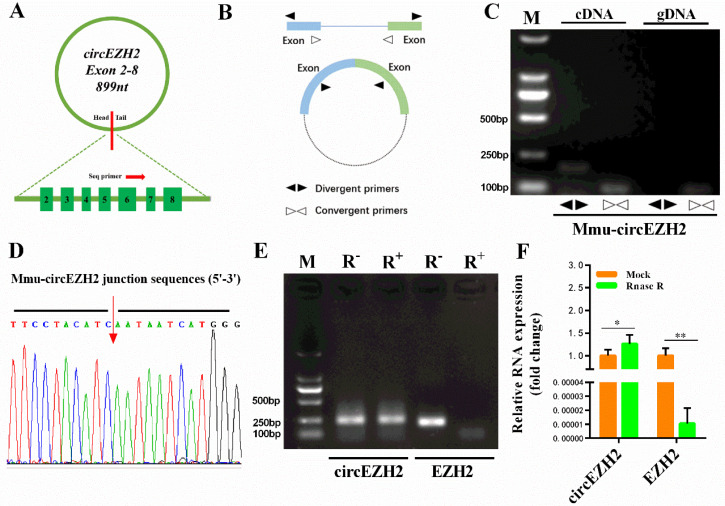
Validation of mouse circEZH2. (**A**) Schematic diagram of circEZH2. (**B**) Schematic diagram of divergent primer and convergent primer design. (**C**) Identification of circEZH2 by divergent primers and convergent primers. Divergent primers amplify circEZH2 in cDNA but not genomic DNA (gDNA). (**D**) Sanger sequencing results using divergent primer pairs. (**E**) RNase R enzyme digestion electrophoresis. (**F**) qRT-PCR analysis of RNA expression. Data represent means ± standard deviation, *, *p* < 0.05; **, *p* < 0.01.

**Figure 2 animals-12-00718-f002:**
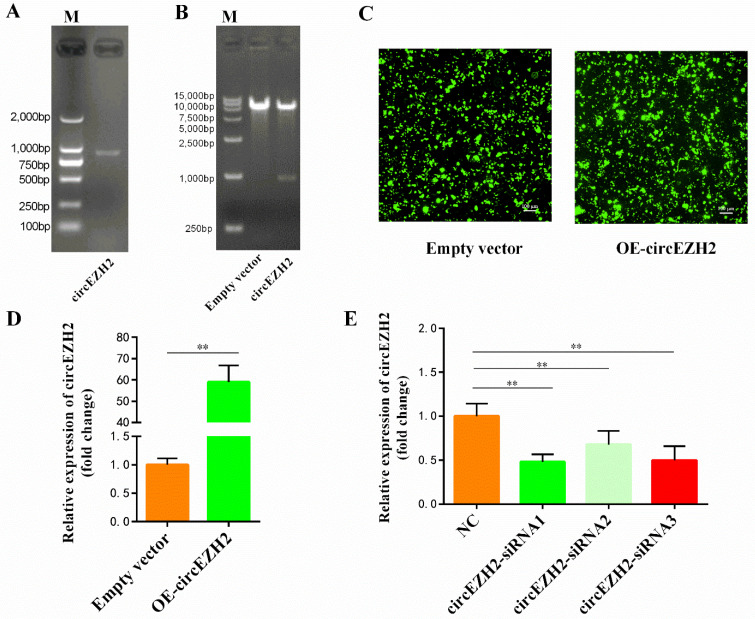
Construction of circEZH2 overexpression vector and interference sequence. (**A**) Agarose electrophoresis of full-length circEZH2. (**B**) Agarose electrophoresis following double enzyme digestion of circEZH2 overexpression vector and empty vector. (**C**) The GFP protein expression tested by fluorescence microscopy. (**D**) qRT-PCR analysis of the expression efficiency of the circEZH2 overexpression vector. (**E**) qRT-PCR analysis of the efficiency of the interference sequences. Data represent means ± standard deviation, **, *p* < 0.01.

**Figure 3 animals-12-00718-f003:**
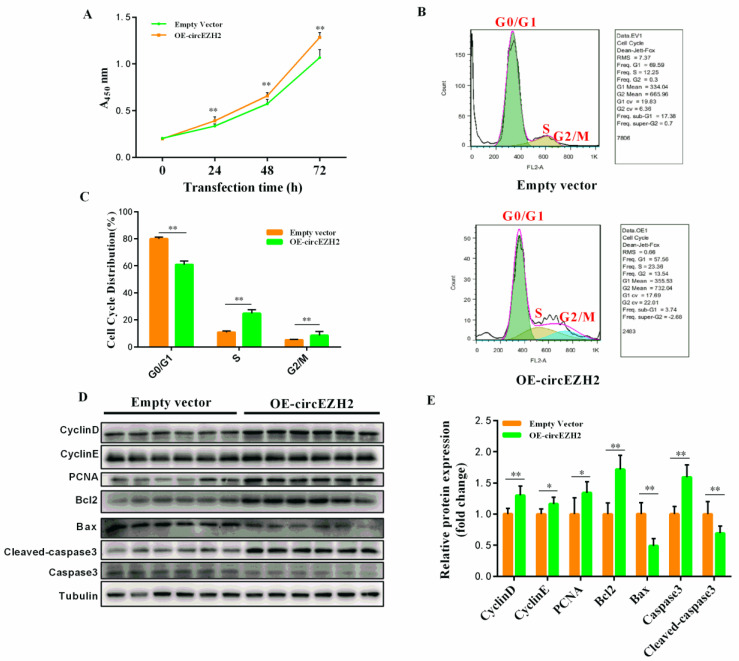
CircEZH2 overexpression vector affects cell proliferation and apoptosis. (**A**) CCK8 analysis of HC11 cell proliferation. (**B**) Flow cytometric analysis of cell cycle distribution. (**C**) Quantitative analysis of cell cycle distribution. (**D**) Western blot analysis of the expression of proliferation- and apoptosis-related proteins. (**E**) Quantitative analysis of the expression of proliferation- and apoptosis-related proteins. Data represent means ± standard deviation, *, *p* < 0.05; **, *p* < 0.01.

**Figure 4 animals-12-00718-f004:**
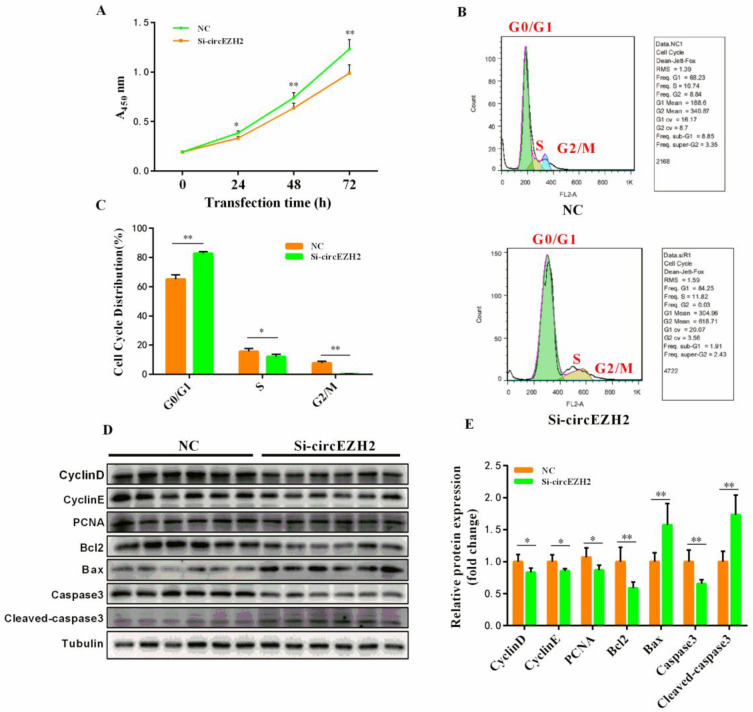
Silencing of circEZH2 expression affects cell proliferation and apoptosis. (**A**) CCK8 analysis of HC11 cell proliferation. (**B**) Flow cytometric analysis of cell cycle distribution (**C**) Quantitative analysis of cell cycle distribution. (**D**) Western blot analysis of the expression of proliferation- and apoptosis-related proteins. (**E**) Quantitative analysis of the expression of proliferation- and apoptosis-related proteins. Data represent means ± standard deviation, *, *p* < 0.05; **, *p* < 0.01.

**Figure 5 animals-12-00718-f005:**
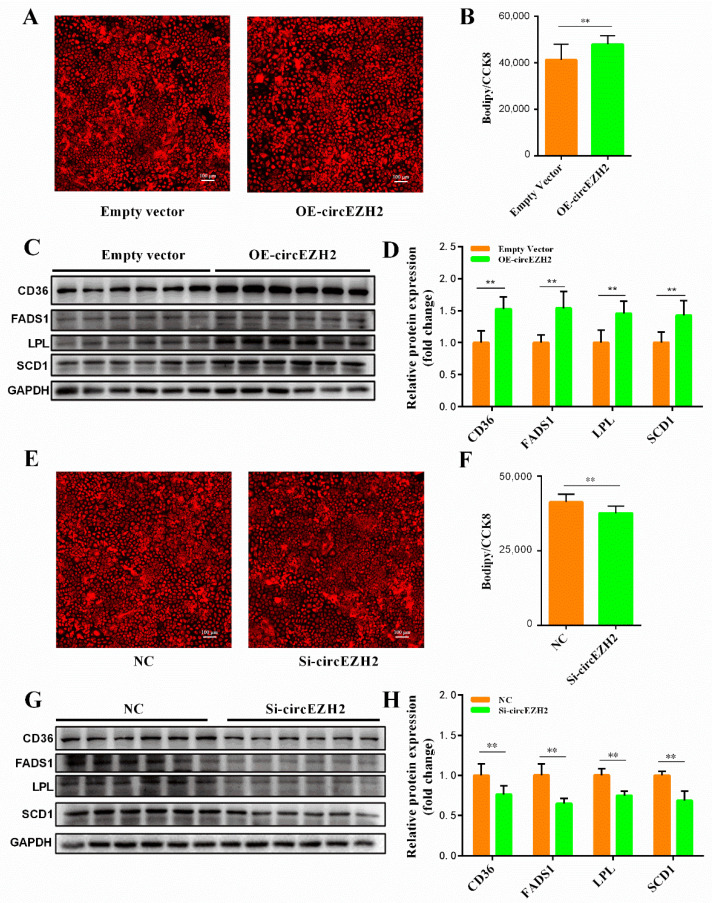
CircEZH2 affects cell lipid metabolism. (**A**,**E**) Fluorescence imaging of fatty acid uptake assay. (**B**,**F**) Quantitative analysis of fatty acid uptake. The value was normalized to CCK8 analysis. (**C**,**G**) Western blot analysis of lipid metabolism-related protein expression. (**D**,**H**) Quantitative analysis of lipid metabolism-related protein expression. Data represent means ± standard deviation; **, *p* < 0.01.

**Figure 6 animals-12-00718-f006:**
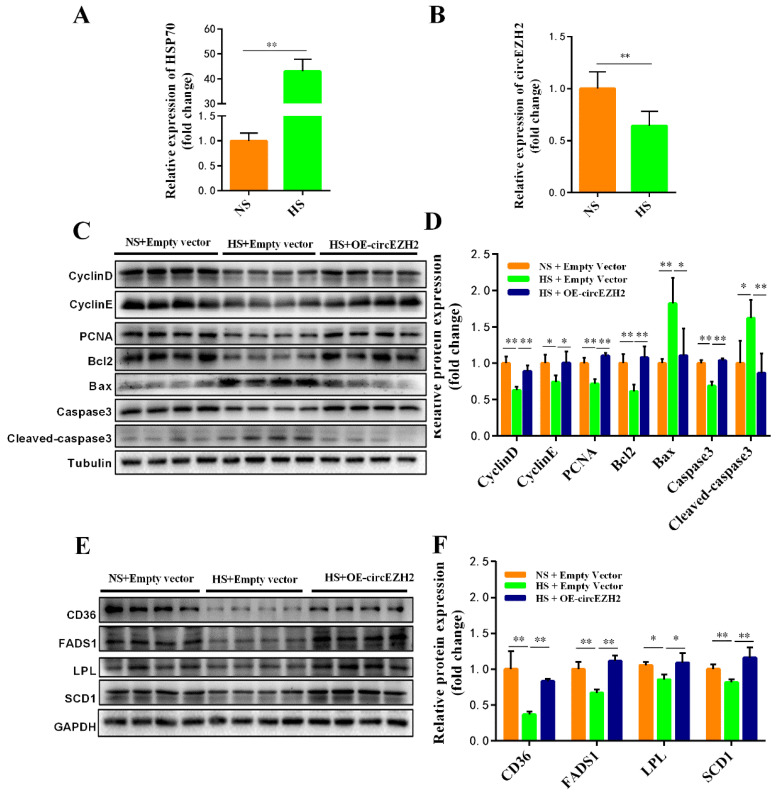
CircEZH2 relieves the negative effects of heat stress. (**A**) qRT-PCR analysis of HSP70 expression. (**B**) qRT-PCR analysis of circEZH2 expression. (**C**,**D**) Western blot analysis of the expression of proliferation- and apoptosis-related proteins under hear stress. (**E**,**F**) Western blot analysis of the expression of lipid metabolism-related proteins under heat stress. Data represent means ± standard deviation, *, *p* < 0.05; **, *p* < 0.01.

**Figure 7 animals-12-00718-f007:**
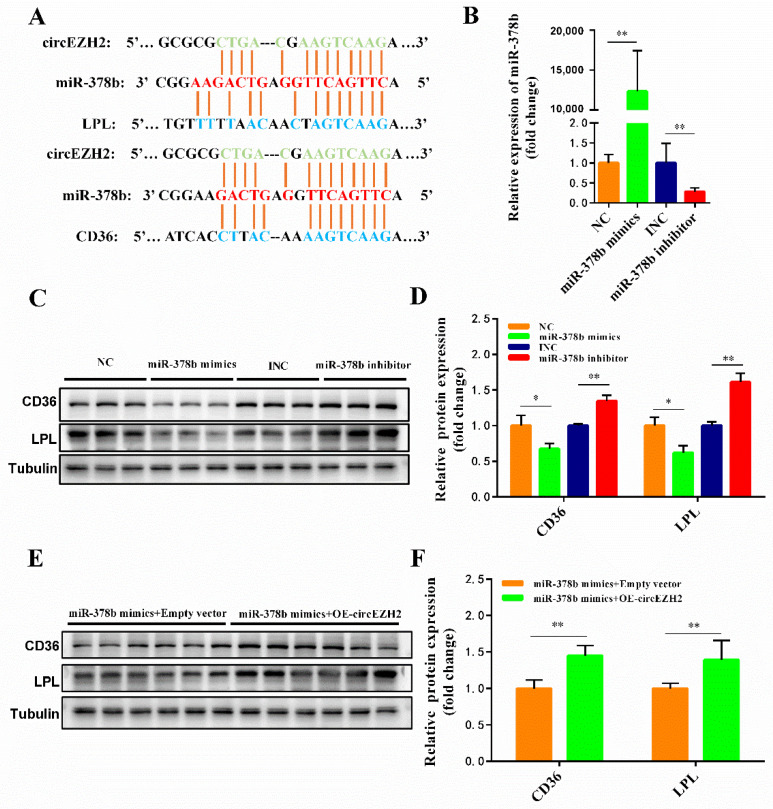
Prediction of circEZH2 competitive regulatory network. (**A**) Schematic diagram of target relationship prediction. (**B**) qRT-PCR analysis of the efficiency of miR-378b mimics and inhibitor. (**C**,**D**) Western blot analysis of the changes in LPL and CD36 protein expression following co-transfection with miR-378b and circEZH2. (**E**,**F**) Western blot analysis of the changes in LPL and CD36 protein expression following co-transfection with or miR-378b and circEZH2. Data represent means ± standard deviation, *, *p* < 0.05; **, *p* < 0.01.

**Figure 8 animals-12-00718-f008:**
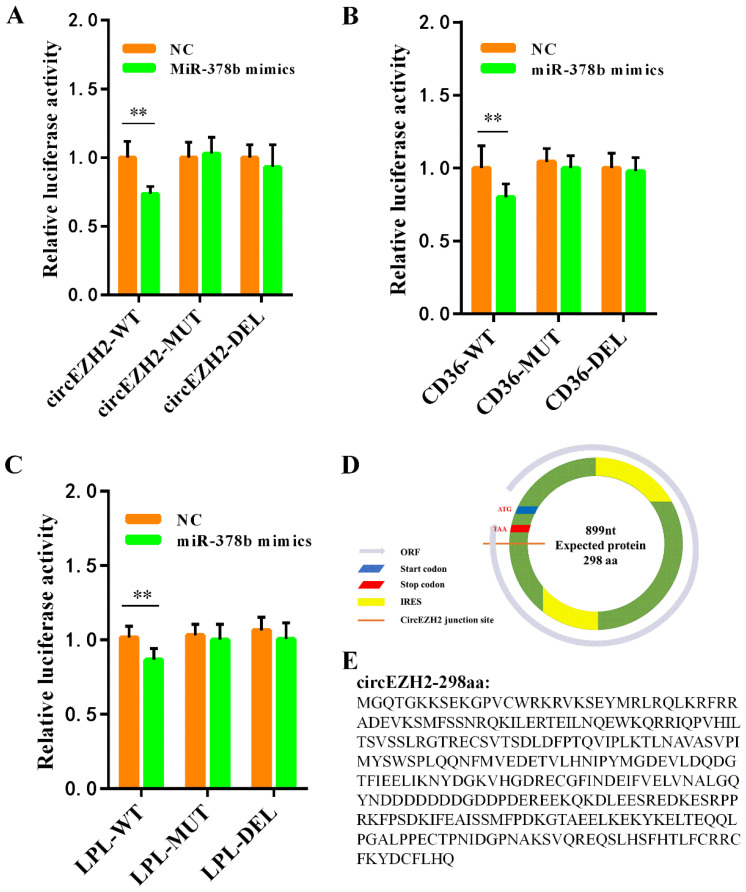
Verification of target relationship and prediction of translation ability. Dual luciferase reporter gene assay verifies the target relationship between miR-378b and circEZH2 (**A**), CD36 (**B**), or LPL (**C**). (**D**) Schematic diagram of circEZH2 translation potential. (**E**) Putative polypeptide sequence encoded by circEZH2. Data represent means ± standard deviation, **, *p* < 0.01.

## Data Availability

All data are contained in the manuscript.

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
