# Peer review of "CircEZH2 Regulates Milk Fat Metabolism through miR-378b Sponge Activity"

_animals, 2022, doi:10.3390/ani12060718_

Round 1

Reviewer 1 Report

The following minor changes can be complied with

Please find the following comments 

  1. Abbreviations are not properly mentioned.
  2. HeLa cells are used in these experiments that are good cancer cells, please explain their significance in current study.
  3. Increase the chromatogram length in figure 1D, with clearly showing exon sequence with chromatogram.
  4. Typo error in lines - 154 and 169
  5.  Need more explanation on figure 1c.

Reviewer 2 Report

Title of Manuscript:  CircEZH2 regulates milk fat metabolism through miR-378b  sponge activity.

By authors: Dongyang Wang, Zhengjiang Zhao, Yiru Shi, Junyi Luo, Ting Chen, Qianyun Xi, Yongliang Zhang and Jiajie  Sun

This manuscript is very good contribution about influence of heat stress on milk production and fat content in milk in dairy cows. This research are addressed on the roles of heat-induced circEZH2 in regulation of milk fat metabolism. CircEZH2 overexpression promoted the uptake of a fluorescent fatty acid (Bodipy) as well as expression of the fatty acid transport-related protein CD36, lipolysis-related  protein LPL, and unsaturated fatty acid metabolism-related proteins FADS1 and SCD1.

This manuscript is original reasarch paper with deeply explanation of complex gene expression in mamocytes during heat stress in dairy cows which decreased  milk production and fat content in milk.

The Key parts of maniscript (Material and methods and Results) are clearly presented.

The manuscript is good and clear written.

The conclusion is in adequate form.

Conclusions:  We found that HC11 cell proliferation is inhibited and apoptosis is promoted following exposure to heat stress. Lipoprotein hydrolysis, fatty acid uptake and unsaturated fatty acid production were also significantly reduced. Thus, we conclude that the heat stress-related circEZH2 effectively alleviates the adverse effects of heat stress in HC11 cells  through two ceRNA competitive regulatory networks (circEZH2-miR378b-CD36, cir-449 cEZH2-miR378b-LPL) and one encoded polypeptide to restore cell proliferation, apoptosis and lipid metabolism.

This manuscript is acceptable for publicitation in Journal (Animals) with high Impact factor.

Author Response

Thanks so much.

Reviewer 3 Report

Wang and colleagues build off their previously published work to further explore the role of miR-378 in regulating milk fat metabolism. Overall, the investigation presented in this manuscript is interesting and performed well. I have a few points that should be addressed prior to publication.

1. While reading, I was not sure which miRNA the authors are referring to. Based on their primer sequence for the mimic, it appears to be bta-miR-378b, but this should be clarified early in the manuscript using the appropriate nomenclature for miRNA (see: www.mirbase.org)

2. The "Simply summary" starts off well, but quickly uses jargon and undefined acronyms (e.g., ceRNA). Please rework this to be understandable by a non-expert reader.

3. At the end of the introduction, the authors should be more clear that this study was conducted entirely in mouse cells and that while informative, this study does not provide any direct evidence for "mechanisms that control dairy cow lactation and milk quality." Also applies to lines 375-8 and 435 in the discussion.

4. The title of Figure 1 should be changed as the contents do not show any homology with bovine sequences but instead detail the cloning/circularization of the mouse gene.

5. Further to point 4, can the authors provide evidence that EZH2 is circular when over expressed by the plasmid? Most of the results could be replicated with a linear EZH2. This is apparent in the luciferase assays where EZH2 cannot possibly be circularized.

6. In Figure 2 panels D and E, it would be more informative to qPCR data relative to a housekeeper gene rather than normalized to controls to give a sense of the level at which these transcripts are expressed in the cell.

7. Line 214, please interpret the OD value for the reader (i.e., a measure of proliferation) when describing the results.

8. The flow cytometry analysis in Figs 3 and 4 has not been performed correctly. In both panels D the gating (quadrants) has been changed between the two treatment groups. Further, the empty vector bars in Fig 3C do not appear to sum to 100%.

9. Lines 256-7 the authors do not mention controlling for cell proliferation (which was demonstrated to be higher in EZH2 group) although Fig 5. suggests that this was normalized to CCK8. Please clarify in the text.

10. Further to 9, the method of quantification of FA uptake should be explained in more detail. For instance, how many fields were used and how many cells per field were measured.

11. Figure 5G states that the control is Tubulin, but I suspect it is GAPDH (as labelled in the full blot images). The miR-378b inhibitor in Fig 7 is not referred to at all in the text or the legend.

12. I appreciated what the authors were trying to do with the ORF analysis, but my view is that it detracts from the whole manuscript (particularly as circRNA are introduced as non-coding RNAs) and given a limited bioinformatic analysis (i.e., no confirmation of the predicted peptide) it adds little to the conclusions of the paper.

Reviewer 4 Report

General remarks -

In this study, the authors investigated the function of circRNA in fat deposition of milk by using mouse HC11 cell line and recombinant expression plasmids. The logic behind this investigation is result of the group former study and its role in fat synthesis in milk.. Hence, it seems reasonable to follow this in the current study.

Major commnet -

While the authors used the HC11 mouse mammary epithelial cell line, most of the discussion extrapolated conclusions on heat stress in cows. The connection is understood, but is too speculative without studying similar effects using Cow mammary tissue.

General comments

-Review the abbreviation list at the end of the article, I believe that many abbreviations used are missing or need to be listed in full in the text and then listed here..

Gene names should be listed and italics and need to be listed in full before using their abbreviations.

  • GeneID number data will be helpful as well (e.g. from the Mouse Genome Database @ http://www.informatics.jax.org/)

Figures should be stand alone, hence you need to explain abbreviations in their legends.

There are too many times that you refer to your previous study results. Please concise it only to where it is necessary to understand the reasoning behind this study and as comparison point to the current study results.

Abstract

  • Need to give full names of genes when first introduced and mark genes in italics
  • missing info on why this function is important and which methods used.

Introduction -

Line 50: insert "and" before "sows"

Line 55: Give full name of the abbreviation "CircEZH2"

Rephrase the last 3 lines of introduction as hypothesis/aim instead of as summary of your results.

Methods

General comments -

The reasoning behind why miR378b was chosen to be used in the vector you created is only explained towards the end of your discussion section. Consider bring this forward.

Until figure 3 legend it was not epxlained why you selected to check the set of genes in your RT-qPCR assay (i.e. because these are proliferation- and apoptosis-related proteins). I suggest to highlight this earlier in the text/methods

  • Need reference to the 2-ΔΔCt method calculation +/- short description of this step.
  • Lines 88-89: "and restriction enzymes XbaI 88 and XhoI were added at both ends of the sequence" - did you mean that the matching sequence of these restriction enzymes was added?
  • Line 93 - explain the abbreviation of "mmu-miR" and what is "mimics"?
  • Line 97 - "National Collection..." of which Country?
  • Line 104 - From which companies are the "...insulin, 100 nM dexamethasone and 5 μg/mL prolactin"?
  • Western blot - Define full name for RIPA and BCA; details of the anti-rabbit anti mouse Ab used needed (e..g company ordered from);  Full name of genes needed;  How long the membranes were incubated with the secondary AB? ; Version of Image J software?
  • Line 124 - "transfected for different periods of time" - specify the different periods.
  • Line 125 - verb missing in "Subsequently, the cells with CCK-8 reagent  for about 2 hours"?
  • Line 132 and 138 - what was the % of the "ice-cold PBS" and "pre-chilled PBS solution"?
  • Line 145 - "cultured normally" - explain what are the "normal" conditions.
  • Line 153 - "DMEM" - abbreviation for?
  • Line 169 - revise the structure of the middle sentence - should it be deleted or something missing?
  • Statistics methods - Not clear when you used the threshold of P < 0.05 and when of P < 0.01?

Results

It is very hard to view the bands in Fig 1C - this may work better if you can use an image with better resolution in the next version?

It is not clear how Figure 1 prove homology to the cow sequence - seems that it only shows results from your mouse samples?

Some of the description in Construction of the circEZH2 overexpression vector section 3.2 will be fit better to methods. Did you explained what is "siRNA", GPF and "NC" earlier?

Lines 251-256 and 333-344 will fit better in the discussion section

Line 334- what is "ceRNA"? (which you late explained in lines 418-419...)

Discussion

The first two paragraphs fir to the introduction better, or need to be shorten if used in discussion.

Line 576 - "we investigated the changes in the expression of circEZH2 in dairy cow mam-376 many tissues"... I may have missed something, but it seems that you used mouse cell line and not cow tissue?

Line 436 - Please explain why "circRNAs function as miRNA sponges"?

Round 2

Reviewer 3 Report

I would like to thank the authors for their comprehensive edits to the manuscript.

This manuscript is a resubmission of an earlier submission. The following is a list of the peer review reports and author responses from that submission.